# Orchestrating Heterogeneous Experts: A Scalable MoE Framework with Anisotropy-Preserving Fusion

Ye Liu
ly490081@alibaba-inc.com
Institute of Intelligent Technology, Alibaba International
Digital Commerce Group
Hang Zhou, China

Wuji Chen[*]
wuji.cwj@lazada.com
Institute of Intelligent Technology, Alibaba International
Digital Commerce Group
Hang Zhou, China

Xu Chen[*]
wenmo.cx@alibaba-inc.com
Institute of Intelligent Technology, Alibaba International
Digital Commerce Group
Hang Zhou, China

Mang Li[†]
mang.ll@lazada.com
Institute of Intelligent Technology, Alibaba International
Digital Commerce Group
Hang Zhou, China

## Abstract

In cross-border e-commerce, search relevance modeling faces the dual challenge of extreme linguistic diversity and fine-grained semantic nuances. Existing approaches typically rely on scaling up a single monolithic Large Language Model (LLM). However, our empirical analysis reveals that single models suffer from uneven capability distributions across regions. For example, excelling in English while underperforming in specific Southeast Asian languages. In this work, we shift the paradigm from scaling a single model to orchestrating heterogeneous experts. We propose a scalable Coarse-grained Mixture-of-Experts (MoE) framework that leverages the inherent complementarity of distinct open-source LLMs (e.g., Qwen, Gemma) without expensive pre-training. Unlike standard token-level MoE, our framework dynamically routes entire queries to specialized experts and, crucially, employs an Information-Preserving Concatenation Fusion strategy. We theoretically posit that preserving the distinct embedding manifolds of heterogeneous experts—rather than compressing them via weighted averaging—is essential for capturing complex relevance signals in a multi-model latent space. On datasets spanning six Southeast Asian markets, our MoE improves AUC by 0.72 percentage points over a dense baseline with the same active parameters. Meanwhile, the optimized pipeline achieves 13.72 queries per second (QPS), a 9% throughput improvement.

## CCS Concepts

• **Applied computing** → **Online shopping**; • **Computing methodologies** → *Natural language generation*; • **Information systems** → **Query intent**.

## Keywords

E-commerce Retrieval, Search Relevance, Large language models

## 1 Introduction

In the rapidly expanding landscape of cross-border e-commerce, search relevance modeling serves as the linchpin for both user

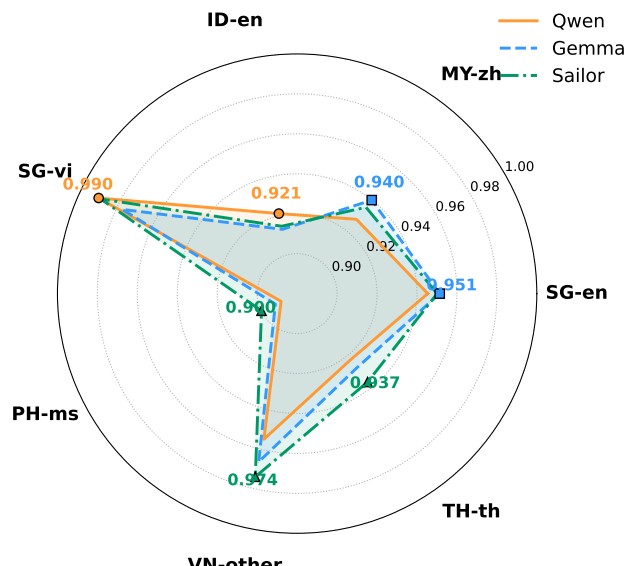

**Figure 1: Radar chart showing the AUC performance of three base models across multiple country–language tasks. Different models exhibit distinct capability skews (e.g., Qwen excels in ID-en, Sailor in TH-th), highlighting the potential for complementarity through a Mixture-of-Experts approach.**

experience and platform revenue. Unlike standard web search, e-commerce queries in Southeast Asian markets are characterized by extreme brevity, intent ambiguity, and, crucially, a highly fragmented linguistic distribution. A single platform must often process English queries requiring deep semantic reasoning while simultaneously handling local languages (e.g., Thai, Indonesian) where syntactic exactness and local entity recognition are paramount.

Traditionally, the industry has relied on scaling up monolithic Pre-trained Language Models (PLMs), such as BERT-based encoders or larger unified LLMs, to address this diversity. However, this "one-model-fits-all" paradigm faces a steep curve of diminishing returns. Training a single massive model to master all linguistic nuances

---

[*]Both authors contributed equally to this research.
[†]Corresponding author.

and domain-specific knowledge (e.g., local brands, slang) is not only computationally prohibitive but often leads to the "curse of multilingualism," where optimizing for high-resource languages can inadvertently degrade performance on low-resource ones due to capacity dilution [1].

Our empirical analysis (Figure 1) shows that even with identical fine-tuning data/objectives, different LLMs (Gemma2-9B[8], Qwen2.5-14B[9], Sailor2-20B[2]) vary in performance across languages/regions. For example, Sailor2-20B (larger scale, Southeast Asian languages pre-training) does not outperform others in some languages, while Qwen2.5-14B excels in Indonesian–English and Singapore–Vietnamese. This suggests that differences in pre-training corpora, vocabulary design, and language coverage lead each model to internalize distinct regional language capabilities, which remain complementary even under identical fine-tuning. This observation drives our core hypothesis: Instead of forcing a single model to learn everything, can we orchestrate a mixture of specialized, heterogeneous experts to achieve a Pareto-optimal balance between performance and cost?

To leverage this complementarity, we propose a lightweight LLM-based Mixture-of-Experts (MoE) framework that integrates fine-tuned LLMs as independent experts without additional fine-tuning. The routing module dynamically assigns each input to the most suitable experts, and their representations are fused at the embedding level, leveraging both the improvements within each expert and the complementary strengths across models. We explore various routing strategies, including rule-based, pseudo-label, and end-to-end (soft/hard) approaches, and find that **End-to-end Hard Routing** delivers the best overall balance.

Another key design decision in our framework is the embedded fusion strategy, which has often been overlooked in previous work. Traditional ensemble methods typically employ Weighted average (Scalar Mixing) to combine expert outputs. However, we argue that simply averaging embeddings from heterogeneous architectures (with distinct latent manifolds and tokenizer spaces) causes destructive interference. Since the basis vectors of Expert A and Expert B are not naturally aligned, a linear combination often collapses distinct semantic signals into noise. To address this, we introduce an Information-Preserving Concatenation mechanism. By preserving the full dimensionality of expert outputs in a concatenated space, we allow the downstream classifier to learn a non-linear decision boundary, effectively acting as a discriminator that learns which expert to trust for specific feature subspaces (e.g., relying on Qwen for syntax and Gemma for attributes).

Furthermore, to ensure industrial viability, we implement an asynchronous batch inference pipeline, decoupling heavy expert computation from real-time routing logic. This allows us to deploy powerful LLM experts within strict latency constraints.

Our main contributions are summarized as follows.

- **Architectural Insight**: We identify the distinct complementarity between heterogeneous LLMs in cross-border e-commerce and propose a framework to exploit this diversity without expensive continued pre-training.
- **Manifold-Preserving Fusion**: We theoretically and empirically demonstrate that concatenation-based fusion outperforms traditional scalar mixing by preserving the distinct

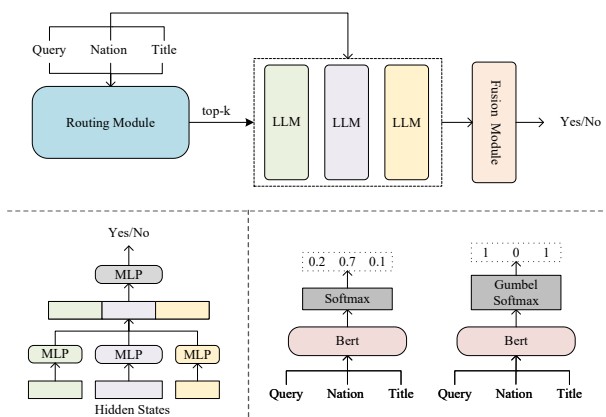

**Figure 2: Overview of the proposed sparsely-activated Mixture-of-Experts (MoE) framework for multilingual e-commerce search. The routing module dynamically selects top-k LLM experts based on query, nation, and item title; their hidden representations are projected and concatenated in the fusion module for relevance prediction. The bottom part shows the architectures of the routing and fusion modules, including soft and hard routing strategies.**

geometric structures of heterogeneous expert embeddings, resolving the issue of feature misalignment.
- **Industrial Effectiveness**: Validated on a large-scale real-world dataset spanning six markets, our approach achieves a 0.72 percentage point AUC gain over parameter-equivalent dense baselines and significantly improves throughput via an optimized asynchronous pipeline.

## 2 Methodology

### 2.1 Framework Overview

To address the challenges of linguistic diversity and semantic complexity in cross-border e-commerce, we propose a scalable **Coarse-grained Mixture-of-Experts (MoE)** framework. Unlike standard token-level MoE architectures that activate experts for each token generation step, our framework operates at the request level, dynamically routing entire queries to specialized Large Language Models (LLMs) and fusing their semantic representations.

**Notation**. We use lowercase bold letters (e.g., $\mathbf{x}, \mathbf{h}$) to denote vectors and uppercase bold letters (e.g., $\mathbf{W}$) for matrices. Sets are denoted by calligraphic letters (e.g., $\mathcal{S}$). $N$ denotes the total number of experts and $k$ denotes the number of active experts per query.

**Architecture Overview.** As illustrated in Figure 2, the inference pipeline consists of three decoupled stages designed to balance effectiveness and efficiency:

(1) **Dynamic Routing**: A lightweight routing module analyzes the input tuple (q,t,c) and selects a sparse subset of top-$k$ experts, $\mathcal{S}_k \subset \{1, \ldots, N\}$, where $k \ll N$, from a set of $N$ heterogeneous LLM experts. This sparsity ensures that computational resources are allocated only to the most suitable models for a given region or language.

(2) **Expert Inference**: The selected experts process the input in parallel. For each selected expert $i \in \mathcal{S}_k$, we extract the last-token hidden state $\mathbf{h}_i$ from its final transformer layer. This vector encapsulates the expert's semantic understanding of the query-item pair.

(3) **Representation Fusion**: The distinct hidden states $\{h_i | i \in S_k\}$ are aggregated by a fusion module FF into a unified relevance score. A critical challenge here is integrating representations from heterogeneous feature spaces, which we address via a manifold-preserving strategy.

The overall training objective minimizes the discrepancy between $\hat{y}$ and ground truth $y$, regularized by a load-balancing term to prevent routing collapse. The details of the routing and fusion mechanisms are described in Section 2.2 and Section 2.3, respectively

## 2.2 Dynamic Sparse Routing

The routing module balances predictive performance and computational efficiency. We considered four strategies:

**Rule-based routing** assigns each query to a single expert based on handcrafted criteria (e.g., language). It is non-parametric, incurs no additional cost, but is rigid and limited to top-1 selection.

**Pseudo-label routing** uses a two-stage pipeline: first, LLMs infer pseudo-labels for the best-performing expert per sample; then a lightweight router is trained to predict these labels, followed by training the fusion module to align models.

**End-to-end soft routing** outputs a probability distribution over all $N$ experts and applies an entropy minimization regularizer to encourage sparsity. This encourages sharper expert selection and model specialization. During inference, only experts exceeding a threshold are activated. Soft routing may suffer from threshold sensitivity and train–inference mismatch.

**End-to-end hard routing** directly selects the top-$k$ experts (e.g., $k = 2$) and incorporates a load balancing loss to prevent routing collapse. Differences in pre-training corpora and architectures produce heterogeneous hidden representations across experts. Early in training, the fusion module may overfit to the easiest expert, causing vanishing gradients for the router. To mitigate this, we adopt a load balancing loss [4]:

$$L_{\text{LB}} = N \cdot \sum_{i=1}^{N} p_i \cdot \bar{p}_i, \tag{1}$$

where $p_i$ is the fraction of samples routed to expert $i$, and $\bar{p}_i = 1/N$ is the ideal uniform usage. This stabilizes training, ensures diverse expert utilization, and maintains consistency between training and inference. Hard routing also provides predictable latency and avoids threshold tuning, which we adopt as the default.

## 2.3 Manifold-Preserving Fusion Strategy

A crucial challenge in utilizing heterogeneous experts lies in the aggregation of their output. While standard MoE architectures typically employ Scalar Mixing (Weighted Averaging), defined as $\mathbf{h}_{mix} = \sum w_k \mathbf{h_k}$, we demonstrate that this approach is theoretically suboptimal for aggregating heterogeneous LLMs.

**The Misalignment Problem and Destructive Interference**. Scalar mixing relies on the strong assumption that the latent spaces

of all experts are isomorphic and semantically aligned—i.e., the $i$-th dimension of Expert A's embedding vector encodes the same semantic feature as the $i$-th dimension of Expert B.

However, for heterogeneous models (e.g., Qwen vs. Gemma) pre-trained on distinct corpora with different tokenizers and architectures, the latent manifolds are topologically distinct. Even if projected to the same dimensionality, the basis vectors are not naturally aligned. Consequently, performing a linear combination on these unaligned vectors leads to destructive interference, where a distinct feature signal in one expert (e.g., a specific syntax pattern) may be cancelled out or diluted by noise from another expert in the same dimension. This results in a "feature collapse" where fine-grained semantic information is lost during aggregation.

**Manifold Preservation via Concatenation.** To address this, we propose a Manifold-Preserving Fusion strategy. Instead of collapsing the representations into a shared lower-dimensional space, we construct a joint representation space by concatenating the selected expert embeddings.

First, for each selected expert $i$, we extract its final transformer layer's last-token hidden state $\mathbf{h}_i \in \mathbb{R}^{d_i}$ and project it to a normalized space:

$$\mathbf{h}_i' = \mathbf{W}_i \mathbf{h_i} + \mathbf{b_i}, \mathbf{h}_i' \in \mathbb{R}^d \tag{2}$$

Where $\mathbf{W}_i \in \mathbb{R}^{d \times d_i}$ projects the expert-specific dimension $d_i$ to a shared dimension $d$.

Then, we form the joint representation $\mathbf{z}$ by concatenating the projected embeddings:

$$\mathbf{z} = [\mathbf{h}_1'; \mathbf{h}_2'; \ldots ; \mathbf{h}_k'] \in \mathbb{R}^{k \times d} \tag{3}$$

Geometrically, this operation preserves the intrinsic manifold structure of each expert within independent subspaces of $\mathbf{z}$, avoiding the destructive interference inherent in averaging.

**Non-linear Decision Boundary.** To extract relevant signals from this preserved high-dimensional space, we employ a lightweight classifier (MLP) as a non-linear discriminator to predict the relevance score $\hat{y}$:

$$\hat{y} = \sigma(\mathbf{W_c} \cdot \text{ReLU}(\mathbf{W}_p \mathbf{z} + \mathbf{b}_p) + \mathbf{b}_c) \tag{4}$$

Unlike a simple dot-product attention mechanism used in scalar mixing, this learnable MLP allows the model to capture cross-expert interactions (e.g., "if Expert A detects high semantic relevance AND Expert B detects brand mismatch, then label as negative"). This enables the system to effectively select the most reliable signal source for the specific query instance without suffering information loss.

**Training Loss**. The model is trained end-to-end. We employ the standard Cross-Entropy loss for the relevance classification task:

$$L_{CE} = -\frac{1}{|\mathcal{B}|} \sum_{(q,t,y) \in \mathcal{B}} (y \log \hat{y} + (1 - y) \log(1 - \hat{y})) \tag{5}$$

where $\mathcal{B}$ is the training batch. The final objective combines this with the load balancing loss $L_{LB}$ (defined in Eq. 1) to ensure expert diversity:

$$\mathcal{L}total = L_{CE} + \lambda L_{LB} \tag{6}$$

where $\lambda$ is a hyperparameter that balances task performance and expert utilization.

**Table 1: Overall and per-market AUC (%) and QPS for all baselines and our proposed MoE. Bold indicates the best, underline the second-best. Our Proposed MoE achieves the best relevance performance across most markets while maintaining competitive throughput compared to dense baselines. Note that "Full Fusion" lags behind MoE, validating the effectiveness of sparse routing.**

| Model | AUC (↑) | | | | | | | QPS (↑) |
|---|---|---|---|---|---|---|---|---|
| | Overall | ID | MY | PH | SG | TH | VN | |
| Qwen2.5-14B | 91.40 | 86.08 | 92.44 | 88.99 | 93.62 | 93.82 | 93.41 | 28.34 |
| Gemma2-9B | 91.40 | 86.35 | **93.71** | 89.49 | 93.09 | 94.00 | 92.94 | **36.76** |
| Qwen+Gemma Fusion | 91.88 | 86.76 | 93.47 | 89.68 | 93.61 | 94.27 | 93.74 | 16.01 |
| Sailor2-20B | 91.82 | 87.63 | 92.40 | 89.24 | 93.14 | 94.65 | 93.64 | 21.90 |
| Qwen2.5-32B | 91.71 | 86.43 | 92.57 | 89.58 | **93.83** | 94.08 | 93.55 | 12.55 |
| Full Fusion (Concat) | 92.41 | 87.89 | 93.70 | 90.25 | 93.75 | 94.94 | **94.13** | 6.94 |
| **Proposed MoE** | **92.49** | **88.33** | 93.56 | **90.45** | 93.62 | **95.00** | 94.10 | 13.72 |

## 2.4 Offline Batch Inference with Resource-Efficient Scheduling

Integrating multiple large LLMs as independent experts introduces substantial computational overhead during inference, particularly in real-world e-commerce environments with high query volumes and strict latency requirements. To address this, we design an offline batch inference pipeline that leverages multi-stream asynchronous execution and resource-efficient scheduling, ensuring high throughput while preserving relevance accuracy.

**Design Principles.** The pipeline is built on three principles: (1) *sparsity-aware computation*, activating only a top-$k$ subset of experts per query; (2) *parallel execution*, overlapping computation and memory operations across experts to exploit GPU-level concurrency; and (3) *scalable scheduling*, dynamically allocating resources to balance load among heterogeneous experts.

**Three-Stage Workflow.** Inference proceeds in three decoupled stages. **(1) Bulk Routing Stage:** The router processes queries in large batches and selects the top-$k$ experts for each query based on query text, region, and item metadata. By precomputing or caching routing decisions where applicable, this stage introduces minimal overhead and enables sparse activation of experts, which reduces memory and compute requirements. **(2) Expert-Specific Batch Inference Stage:** Queries assigned to the same expert are grouped into batches for parallel execution. Each expert independently performs forward propagation on its batch, and multiple experts can run concurrently on separate devices or asynchronously on the same device. This stage exploits GPU parallelism and allows dynamic resource allocation, where faster experts can assist slower ones, improving cluster-level utilization and overall throughput. **(3) Late-Stage Fusion:** After expert inference, the hidden states are aggregated and fused according to the selected fusion strategy (e.g., projection and concatenation). This stage involves only lightweight linear transformations and a small classifier, and can be efficiently executed on lower-cost devices. The design ensures that heterogeneous representations from different LLMs are preserved and combined without loss of complementary knowledge.

## 3 Experiments

### 3.1 Dataset and Evaluation Metrics

We construct a large-scale multilingual e-commerce relevance dataset, collected from real-world Alibaba Lazada search logs and annotated by professional evaluators, covering six Southeast Asian markets: Indonesia (ID), Malaysia (MY), Philippines (PH), Singapore (SG), Thailand (TH), and Vietnam (VN). Each query–item pair is labeled as *Yes* (relevant) or *No* (irrelevant). The dataset comprises 7.1M training samples, 0.7M validation samples, and 46k test samples. We use the Area Under the ROC Curve (AUC) as the primary effectiveness metric and Queries Per Second (QPS) as the efficiency metric. QPS is measured on an NVIDIA H20 GPU and averaged over 1,000 consecutive batches to ensure stable evaluation.

### 3.2 Experimental Settings

Base models—Qwen2.5-14B, Gemma2-9B, Sailor2-20B, and Qwen2.5-32B—are fine-tuned on the same training set using RSLora[5] with rank $r = 256$ and scaling factor $\alpha = 128$. Training is conducted for a single epoch with batch size 4 and learning rate $5 \times 10^{-6}$, following standard practice for large-scale relevance fine-tuning. The MoE framework, including the router, projection layers, and fusion classifier, is trained with batch size 128, learning rate $1 \times 10^{-4}$, and $\lambda = 0.01$ in Eq. 6, while all base LLMs remain frozen.

We include Qwen2.5-32B as a strong dense baseline, whose total parameter count (32B) is comparable to the *maximum active parameters* of our MoE framework ( 34B for two experts). This enables a fair comparison between a scaled-up homogeneous model and our sparsely-activated heterogeneous MoE. Other baselines include smaller individual models and a **full fusion (Concat)** variant that concatenates all three experts' embeddings without routing.

### 3.3 Main Results

Table 1 reports overall and per-market AUCs and QPS. Our MoE achieves the highest overall AUC (92.49%), surpassing all single models and the full fusion variant—proving its strong multilingual relevance modeling. Across markets, it integrates expert strengths: best performance in ID/PH/TH, and competitive scores in MY/SG/VN. Notably, it outperforms the full fusion model in several regions despite activating fewer experts, suggesting that full fusion may introduce noise and limit effective use of each base model. In terms

**Table 2: Ablation studies on fusion and routing strategies. Reported metrics are overall AUC (%) and QPS.**

| Method | AUC (%) ↑ | QPS ↑ |
|---|---|---|
| *Fusion strategies (Hard routing)* | | |
| Weighted fusion | 92.27 | 13.72 |
| Concatenation | 92.49 | 13.72 |
| *Routing strategies (Concat fusion)* | | |
| Rule-based | 91.94 | 24.44 |
| Pseudo-label | 91.57 | 34.47 |
| Soft routing | 92.09 | 16.01 |
| Hard routing | 92.49 | 13.72 |
| Serial Hard routing | 92.49 | 6.56 |

of efficiency, the MoE achieves a QPS of 13.72, exceeding the dense Qwen2.5-32B (12.55) and substantially outperforming full fusion (6.94), achieving a favorable relevance-throughput trade-off. We also observe that Sailor2-20B, obtained by extending and further pre-training Qwen2.5-14B, improves substantially over its base model. Nevertheless, even a simple fusion of Qwen and Gemma yields higher overall performance than Sailor2-20B, with much lower pretraining cost. This indicates that combining complementary experts through lightweight fusion can be more effective and efficient than scaling a single model via continued pretraining.

### 3.4 Ablation Studies

To assess the contribution of individual components, we conduct ablation studies on fusion strategies and routing mechanisms. All variants are trained and evaluated under identical settings. Table 2 shows concatenation-based fusion outperforms weighted fusion by 0.22 AUC, as it preserves complementary features without premature compression. Among routing strategies, end-to-end hard routing achieves the highest effectiveness, whereas pseudo-label routing attains the highest QPS but performs poorly because most samples collapse to Gemma during training due to difficulty in learning accurate pseudo labels. Soft routing suffers from a train–inference mismatch, leading to suboptimal performance. We also include a *Serial Hard routing* variant, where selected experts are processed sequentially rather than in parallel. While it achieves the same AUC as hard routing, the QPS drops significantly (from 13.72 to 6.56), demonstrating that our optimized parallel inference pipeline nearly doubles throughput while maintaining identical model outputs. This highlights the effectiveness of the resource-efficient scheduling and multi-stream batch execution in practical deployment.

### 3.5 Geometric Interpretation: Manifold Unfolding and Anisotropy

To intuitively demonstrate the effectiveness of our fusion strategy, we visualize the latent representations of the test set using t-SNE (t-Distributed Stochastic Neighbor Embedding). We randomly sampled 45,000 instances from the test set, balanced between relevant (Class 1, blue) and irrelevant (Class 0, red) samples. The visualization compares the final embedding space generated by our

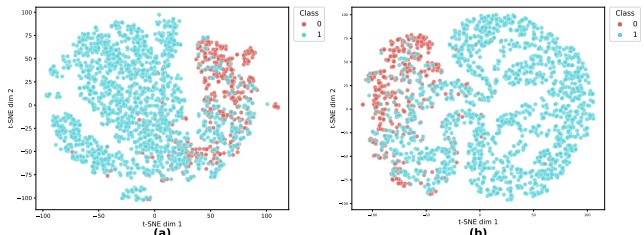
(a)     (b)

**Figure 3: t-SNE Visualization of Embedding Spaces under Different Fusion Strategies. Comparison between (a) Concatenation and (b) Weighted Fusion. The concatenation strategy (a) yields a highly structured manifold with a clear separation gap, effectively avoiding the class entanglement observed in the weighted averaging approach (b).**

Concatenation-based framework (Figure 3 (a)) against the Weighted Fusion baseline (Figure 3(b)). All hyperparameters, including perplexity and learning rate, were kept consistent to ensure a fair topological comparison. This topological separation visualized in Figure 3 (a) serves as the geometric underpinning for the superior AUC scores reported in Table 2. The comparative visualization reveals fundamental topological differences that align with our theoretical hypothesis of Manifold Preservation, providing a geometric explanation for the performance gains reported in Table 2.

**Manifold Unfolding**: Topological Detachment vs. Entanglement The most striking observation in Figure 3 (a) is the emergence of a distinct, isolated subspace for Class 0 samples (visible as the dense red cluster detached on the right). This phenomenon indicates that our method successfully achieves "Manifold Unfolding." By concatenating the representations ($[\mathbf{h}_A; \mathbf{h}_B]$), the system preserves the full dimensionality of the heterogeneous expert spaces. This allows the model to project conflicting samples into orthogonal subspaces, creating a "clean gap" (white space) that makes the data linearly separable for the subsequent MLP.

Beyond topology, these results illustrate the critical role of anisotropy preservation. Pre-trained LLM embeddings are typically anisotropic, encoding semantic information in specific, high-variance directions ("spikes"). When performing weighted averaging on unaligned experts, the dominant semantic direction of one model often acts as noise to the other. This leads to destructive interference, where sharp distinctive features are canceled out, pushing the distribution towards isotropy (a more uniform, spherical distribution). The result is the visual "smearing" seen in Figure 3(b) where the boundaries between classes are softened.

Our Concatenation strategy maintains the orthogonal anisotropy of the experts. The sharp, elongated separation seen in Figure 3(a) proves that the high-variance semantic signals from both models are preserved intact. This allows the downstream classifier to leverage the "sharpest" features from either expert to define a precise decision boundary, effectively resolving the "curse of multilingualism."

### 3.6 Online Deployment

In industrial search advertising systems, upon receiving a user query, a Top-K subset of ads is selected from a vast inventory, following a standard multi-stage pipeline: index → retrieval →

**Table 3: Online performance comparison of the distilled student model (ColBERT) trained by the strongest single teacher (Sailor2) versus our Proposed MoE across six markets. "Bad Ratio" (↓) indicates the percentage of irrelevant results (lower is better), while "Online AUC" (↑) measures ranking accuracy.**

| Model | Bad Ratio (↓) | | | | | | | AUC (↑) |
| | Overall | ID | MY | PH | SG | TH | VN | |
|---|---|---|---|---|---|---|---|---|
| Sailor2-20B | 9.00 | 10.48 | 8.81 | 9.63 | 7.45 | 8.06 | 9.65 | 90.39 |
| **Proposed MoE** | **8.55** | **10.26** | **8.45** | **9.32** | **6.90** | **7.47** | **8.90** | **91.28** |

prerank → rank → auction → rerank. At the retrieval stage, the relevance scores serve dual purposes: filtering candidates for downstream stages and informing dynamic reserve prices in the auction mechanism [3, 7], which balance user experience, platform revenue, and advertiser ROI. Accurate relevance estimation is therefore critical for both result quality and monetization efficacy.

However, industrial-scale retrieval involves tens to hundreds of thousands of candidate items per query. Real-time inference with large language models (LLMs) at this scale is computationally prohibitive due to latency constraints. To leverage LLM semantics without losing efficiency, we use offline knowledge distillation: LLMs generate high-quality relevance labels on a billion-scale query-item corpus to train a compact student model. We select *ColBERT* [6] as the student model due to its strong representational capacity and efficiency in late-interaction architectures. The distilled ColBERT model is deployed online, enabling fine-grained, low-latency relevance scoring that meets the demands of real-time retrieval.

**Online Evaluation**. To verify whether the MoE is a better teacher than single monolithic models, we compare the performance of ColBERT students distilled from different teachers (Qwen, Gemma, Sailor, and our MoE). We focus on two key metrics:

- **Bad Ratio(↓)**: The percentage of irrelevant items in the top retrieval results, evaluated by human experts on a sampled traffic set. Lower is better.
- **Online AUC(↑)**: The ranking capability of the student model on the held-out test set.

**Results**. To validate the effectiveness of our framework in a production setting, we benchmark the MoE-distilled student against a student distilled from Sailor2-20B, which served as the strongest single-model baseline in our offline experiments. As presented in Table 3, the results demonstrate the comprehensive superiority of the MoE teacher:

**Universal Quality Improvement**: The MoE-distilled student achieves a significantly lower Overall Bad Ratio (8.55%) compared to the Sailor baseline (9.00%), representing a relative reduction of 5%. Notably, our approach outperforms Sailor across all six markets, with the most substantial gains observed in Vietnam (9.65% → 8.90%) and Thailand (8.06% → 7.47%). This confirms that the MoE's diverse expert knowledge is effectively transferred, mitigating the performance fluctuations often seen in single models across different languages.

**Enhanced Ranking Accuracy**: Beyond filtering irrelevant results, the MoE teacher also imparts better ranking capabilities. The Online AUC improves from 90.39% to 91.28%, indicating that the student model has successfully learned the fine-grained semantic nuances captured by the manifold-preserving fusion.

This result validates that the MoE teacher provides cleaner, more robust supervision than single models, successfully transferring its fine-grained multilingual understanding to the deployable student model.

## 4 Conclusion

In this work, we propose an LLMs-based Mixture-of-Experts (MoE) framework, which leverages the complementary advantages of LLMs across languages/regions, shifting the paradigm from "scaling a single model" to "coordinating heterogeneous experts." Our empirical analysis highlights a key finding: traditional scalar mixing methods applied to heterogeneous latent manifolds lead to destructive interference. In contrast, our manifold-preserving fusion strategy—relying on feature concatenation and nonlinear discrimination—successfully preserves the unique inductive biases of each expert, enabling the system to achieve Pareto-optimal performance.

Validated on large industrial datasets, our method achieved a 0.72 percentage point improvement in AUC, and the optimized offline pipeline reduced GPU computation time by 35% and increased QPS throughput by 9%. This demonstrates that lightweight coordinated freezing of LLMs is a viable and cost-effective alternative to expensive pre-training. Future work will explore multi-modal integration and automated expert selection.

**Limitations**. Despite its effectiveness, our framework requires loading multiple LLMs into VRAM, which, even with sparse activation, imposes high memory pressure compared to a single small model. Additionally, the coarse-grained routing at the query level may miss token-level nuances.

**Future Work**. We plan to explore: (1) Fine-grained Routing: Investigating token-level or chunk-level expert selection for mixed-language queries. (2) Multi-modal Experts: Integrating image encoders as experts to address queries where visual information is crucial. (3) Model Compression: Applying quantization techniques to further reduce the deployment cost of the expert pool.

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
