# OpenReview forum: "Orchestrating Heterogeneous Experts: A Scalable MoE Framework with Anisotropy-Preserving Fusion"
_ACM.org/TheWebConf/2026/Workshop/TIME — TIME 2026 Poster_

### Meta-Review · Program_Chairs · 2026-02-13

**Recommendation:** Accept (Poster)
**Confidence:** 4

**Metareview:**

This work is submitted after the workshop deadline; hence the review is processed by program chair.

This work explores a very interesting and timely topic. The authors introduce a scalable MoE framework.

Some revision suggestions:

- In abstract, it is suggested not to use "—".

- Fig. 1 font sizes could be improved to be clear. It would be more beneficial to add some descriptions / insights in the caption.

- Pay more attention to punctuations such as "-", "—", "--", etc. They need to be consistent in the whole paper.

- At the end of the intro, it would be better to list 3-4 main contributions to make them clear to readers.

- It would be better to have a notation section detailing the maths symbols and operations used in the paper, e.g., what are scalars, vectors, matrices, etc.

- Line 205-206, what is this $L_{ent}$? this term seems not being used later in the paper? In Eq. (5) what is this $L_{CE}$? These parts appear to be confusing.

- Table 1 caption could add few sentences talking about insights/findings etc.

- Fig 3 legend appears to be unclear (too small font sizes).

- Font sizes presented in the table could be more consistent in the whole paper.

- Conclusion section could be expanded by providing some limitations and future research directions.

All references are checked by PC:

- Ref [3] could cite the published version in JMLR 2022.

- Ref [4] title mismatch issue and venue mismatched issue (AAMAS vs. WSDM).

- Ref [6] could cite their published version in ACM SIGIR.

Overall, the paper is well organised and clean, and this paper can be accepted as TIME 2026 workshop paper on openreview system.

The authors should submit the revised version.

---

### Decision · Program_Chairs · 2026-01-29

**Decision:**

Accept (Poster)

**Comment:**

The authors have provided a revision based on meta review comments.